# Diverse Terpenoids and Their Associated Antifungal Properties from Roots of Different Cultivars of *Chrysanthemum Morifolium* Ramat

**DOI:** 10.3390/molecules25092083

**Published:** 2020-04-29

**Authors:** Kaige Zhang, Yifan Jiang, Hongwei Zhao, Tobias G. Köllner, Sumei Chen, Fadi Chen, Feng Chen

**Affiliations:** 1College of Horticulture, Nanjing Agricultural University, Nanjing 210095, China; 2017104101@njau.edu.cn (K.Z.); chensm@njau.edu.cn (S.C.); chenfd@njau.edu.cn (F.C.); 2College of Plant Protection, Nanjing Agricultural University, Nanjing 210095, China; hzhao@njau.edu.cn; 3Department of Biochemistry, Max Planck Institute for Chemical Ecology, Hans-Knöll Str. 8, 07745 Jena, Germany; koellner@ice.mpg.de; 4Department of Plant Sciences, University of Tennessee, Knoxville, TN 37996, USA

**Keywords:** secondary metabolites, terpenoids, root, antifungal

## Abstract

Roots provide anchorage and enable the absorption of water and micronutrients from the soil for plants. Besides these essential functions, roots are increasingly being recognized as an important organ for the production of diverse secondary metabolites. The goal of this study was to investigate the chemical composition and function of terpenoid secondary metabolites in roots of different cultivars of the popular ornamental plant *Chrysanthemum morifolium* Ramat. Although *C. morifolium* is known for rich production of secondary metabolites in its flower heads and leaves, the diversity of secondary metabolites in roots remains poorly characterized. In this study, 12 cultivars of *C. morifolium* were selected for comparative analysis. From their roots, a total of 20 terpenoids were detected, including four monoterpenes, 15 sesquiterpenes, and one diterpene. The cultivar ‘She Yang Hong Xin Ju’ exhibited the highest concentration of total terpenoids at approximately 730 µg·g^−1^ fresh weight. Most cultivars contained sesquiterpenes as the predominant terpenoids. Of them, (*E*)-β-farnesene was detected in all cultivars. Based on their terpenoid composition, the 12 cultivars were planed into four groups. To gain insights into the function of root secondary metabolites, we performed bioassays to assess their effects on growth of three species of pathogenic fungi: *Fusarium oxysporum*, *Magnaporthe oryzae*, and *Verticillium dahliae*. Significant variability in antifungal activity of the root extracts among different cultivars were observed. The cultivar ‘Xiao Huang Ju’ was the only cultivar that had significant inhibitory effects on all three species of fungi. Our study reveals the diversity of terpenoids in roots of *C. morifolium* and their function as a chemical defense against fungi.

## 1. Introduction

Among the ca. 374,000 known species of land plants, more than 80% are vascular plants with roots [1]. Roots provide anchorage to the plant and enables the absorption of water and dissolved minerals from the soil. Despite the hidden nature, roots interact with diverse organisms in the rhizosphere [2]. Some of these interactions are beneficial to the roots while others are detrimental [3,4]. An increasing body of evidence suggests that the interactions between roots and other organisms are largely mediated by metabolites, particularly secondary metabolites, produced by roots [5]. Despite much progress, in comparison to that in above-ground organs, our knowledge of the chemical composition and diversity of secondary metabolites in roots is very limited. In this study, we were interested in comparative analysis of secondary metabolites produced by the roots of different cultivars of a popular ornamental plant *Chrysanthemum morifolium* Ramat.

*C. morifolium* is one of the oldest ornamental plants. Cultivated in China for thousands of years, it is now an ornamental plant of global importance. Mainly grown as a garden plant or for cut flowers, *C. morifolium* also has other economical values. For example, a number of traditional cultivars of *C. morifolium* are grown for making herbal teas from flower heads [6,7]. During the long history of cultivation, especially with modern breeding techniques, a large number of cultivars have been developed for *C. morifolium*, providing rich genetic resources for this plant. Both flower heads and leaves of *C. morifolium* have been shown to produce diverse secondary metabolites [8,9,10], with terpenoids as one major chemical class [9,10]. Secondary metabolites of *C. morifolium*, like their counterparts in many other plants, are probably involved in diverse interactions between *C. morifolium* plants and the environment [11]. In addition, some of them have been investigated for their health benefits [12] and potential as agrochemicals [13]. In contrast, little is understood about the diversity and functions of secondary metabolites made by the roots of *C. morifolium*.

There are two specific objectives for this study. The first objective is to determine and compare profiles of secondary metabolites from the roots of selected cultivars of *C. morifolium*, which has a tap root system [14]. The entire tap roots of individual plants were harvested and subjected to organic extraction. Extracts were analyzed using gas chromatography-mass spectrometry (GC-MS). The second objective of this study is to associate root secondary metabolites of *C. morifolium* with biological functions. Specifically, root metabolites were tested for their abilities in inhibiting fungal growth. The results from this study will provide new knowledge about both the diversity and function of secondary metabolites from the roots of *C. morifolium*.

## 2. Results and Discussion

### 2.1. Chemical Composition of Terpenoids from C. morifolium Roots

A total of 12 cultivars of *C. morifolium* (Table 1) were analyzed for the chemical composition of apolar secondary metabolites in their roots. For simplicity of presentation, the 12 cultivars were coded as CmR1 to CmR12. A total of 20 terpenoids (Table 2), which included four monoterpenes (C10), 15 sesquiterpenes (C15), and one diterpene (C20), were detected. Besides terpenoids, several non-terpenoid metabolites that are putatively derived from fatty acids were also detected (Appendix A). Generally not considered as secondary metabolites, these compounds were excluded in our further analysis. For the four monoterpenes, α-pinene and β-pinene occurred in five cultivars and α-fenchene and *para*-cymene were detected in four cultivars (Figure 1). Three cultivars, CmR5, CmR11, and CmR12, contained all four monoterpenes. In contrast, monoterpenes were not detected in CmR3, CmR4, CmR6, CmR8, and CmR9 (Figure 1). Sesquiterpenes were the most diverse subclass of terpenoids identified from *C. morifolium* roots and there were large variations in their occurrence and concentrations among cultivars. CmR6 contained 14 sesquiterpenes identified, while CmR7 and CmR10 contained only six sesquiterpenes. For individual sesquiterpenes, (*E*)-β-farnesene appeared to be ubiquitous, while δ-bisabolene and γ-costol were detected only in two cultivars (Figure 1). The single diterpene, which was unidentified (Appendix A), was also ubiquitous (Figure 1). CmR6 had the most diversity of terpenoids with 15 constituents (Figure 1), 14 of which were sesquiterpenes (Appendix A). In contrast, CmR10 was the least diverse, containing eight terpenoids (Figure 1).

Besides the diversity of terpenoids, the concentrations of terpenoids in roots exhibited large variations among the 12 cultivars (Figure 2). For monoterpenes, the total concentrations ranged from 1.32 ± 0.15 to 75.44 ± 5.37 µg·g^−1^ fresh weight (FW) among the seven cultivars, with the highest concentration in CmR12 and the lowest concentration in CmR1 (Figure 2A). For sesquiterpenes, the total concentrations ranged from 190.70 ± 17.79 to 420.31 ± 21.27 µg·g^−1^ FW among the 12 cultivars, with the highest concentration in CmR11 and the lowest concentration in CmR4 (Figure 2B). The concentrations of the single diterpene ranged from 16.99 ± 0.64 to 285.52 ± 39.55 µg·g^−1^ FW among the 12 cultivars, with the highest concentration in CmR10 and the lowest concentration in CmR6 (Figure 2C). For all terpenoids, the concentrations ranged from 249.36 ± 24.97 to 731.58 ± 31.22 µg·g^−1^ FW, with the highest concentration in CmR11 and lowest concentration in CmR4 (Figure 2D).

### 2.2. Grouping of 12 Cultivars of C. morifolium Based on Principal Component Analysis

To establish the relationship of the *C. morifolium* cultivars according to their chemical composition of terpenoids, principal component analysis (PCA) was performed. The 12 cultivars were positioned in the two-dimensional space with the horizontal axis explaining 36.33% of the total variance and the vertical axis explaining a further 21.61% of the total variance (R^2^ = 0.853, Q^2^ = −0.182) (Figure 3). The 12 cultivars were classified into four groups (Group I to Group IV).

Group I contained a single cultivar CmR3. This cultivar was featured with high concentrations of several sesquiterpenes, including silphinene (19.55 ± 0.62 µg·g^−1^ FW), modephene (20.73 ± 0.51 µg·g^−1^ FW), α-isocomene (80.87 ± 2.33 µg·g^−1^ FW), β-isocomene (27.87 ± 0.76 µg·g^−1^ FW) and β-copaene (12.74 ± 0.37 µg·g^−1^ FW).

Group II consisted of three cultivars: CmR5, CmR11, and CmR12. The roots of these three cultivars all contained high concentrations of three monoterpenes, including α-pinene (15.34 ± 1.77 µg·g^−1^ FW, 14.70 ± 2.23 µg·g^−1^ FW, 16.90 ± 2.28 µg·g^−1^ FW, respectively), α-fenchene (27.64 ± 2.29 µg·g^−1^ FW, 32.04 ± 4.12 µg·g^−1^ FW, 33.95 ± 2.68 µg·g^−1^ FW, respectively) and β-pinene (17.84 ± 2.26 µg·g^−1^ FW, 17.14 ± 2.19 µg·g^−1^ FW, 19.45 ± 2.34 µg·g^−1^ FW, respectively). In addition, one sesquiterpene, α-longipinene, occurred in high concentrations in this group with 17.13 ± 1.27, 23.01 ± 2.76, and 18.38 ± 1.91 µg·g^−1^ FW in the roots of CmR5, CmR11, and CmR12, respectively.

Group III was the largest group. It contained six cultivars: CmR1, CmR2, CmR4, CmR7, CmR8, and CmR10. One feature of this group was the partial and full absence of three monoterpenes (α-pinene, α-fenchene, and β-pinene) and four sesquiterpenes (α-longipinene, modephene, α-isocomene, and β-isocomene).

Group IV contained two cultivars CmR6 and CmR9. Both CmR6 and CmR9 contained high concentrations of four sesquiterpenes in their roots, including silphinene (8.52 ± 0.31 and 5.24 ± 0.71 µg·g^−1^ FW, respectively), modephene (9.41 ± 0.30 and 4.99 ± 0.52 µg·g^−1^ FW, respectively), α-isocomene (41.03 ± 1.26 and 22.98 ± 2.99 µg·g^−1^ FW, respectively) and β-isocomene (12.5 0 ± 0.44 and 7.54 ± 0.92 µg·g^−1^ FW, respectively).

### 2.3. Antifungal Activity of C. morifolium Root Extracts

Given the important role of many terpenoids, serving as a chemical defense for many plants [15,16,17], here we evaluated the effects of terpenoid extracts from the roots of 12 cultivars of *C. morifolium* on the growth of three species of pathogenic fungi. The first fungus to be tested was *Fusarium oxysporum*, which is a fungal pathogen naturally occurring to *C. morifolium*, particularly infecting roots and stems [18,19]. Three cultivars, CmR2, CmR5, and CmR7, showed significant inhibition of the growth of *F. oxysporum* (Figure 4A). Of them, CmR2 showed the strongest inhibition effect.

*Magnaporthe oryzae* was the second pathogen to be tested. It is best known as the causative agent of rice blast disease [20]. A previous study showed that the monoterpene limonene can act as a chemical defense against *M. oryzae* [21]. Among the 12 cultivars tested, root extracts of the three cultivars, CmR4, CmR6, and CmR7, showed significant inhibition of the growth of *M. oryzae*, with CmR6 being most effective (Figure 4B).

The third fungus to be tested was *Verticillium dahliae*. As a soil-borne pathogen, *V. dahliae* can cause diseases in a wide range of crop plants, especially those in the Solanaceae family [22]. Significant inhibitory effects on the growth of *V. dahliae* were found with the root extracts of nine cultivars, including CmR2, CmR4, CmR5, CmR6, CmR7, CmR8, CmR9, CmR10, and CmR12 (Figure 4C). Among the 12 cultivars, CmR7 was the only one that exhibited a significant inhibitory effect on the growth of all three species of fungi tested (Figure 4D). It was also notable that none of the 12 cultivars had a significant effect on promoting the growth of any of the three fungi at the concentrations tested (Figure 4).

### 2.4. Partial Least Squares Analysis Based on the Antifungal Effect

In an attempt to assess the relationship between individual terpenoid compounds and fungal growth inhibition, we performed partial least squares (PLS) regression analysis (Figure 5). The model for *F. oxysporum* (Figure 5A) was statistically insignificant (R^2^X = 0.697, R^2^Y = 0.294, Q^2^ = −0.101). In contrast, the analysis for *M. oryzae* (Figure 5B) led to a model with acceptable statistics (R^2^X = 0.852, R^2^Y = 0.693, Q^2^ = 0.513), in which neollocimene (compound 15) exhibited the strongest correlation to growth inhibition (Figure 5B). The analysis for *V. dahliae* (Figure 5C) also led to a statistically insignificant model (R^2^X = 0.739, R^2^Y = 0.285, Q^2^ = −0.210).

## 3. Materials and Methods

### 3.1. Plants Material

All twelve cultivars of *C. morifolium* (Table 1) used in this study are popular varieties with a long cultivation history in China. They are maintained in the Chrysanthemum Germplasm Resource, Preservation Center, Nanjing Agricultural University, China (118°98′ N, 32°07′ E). For all the cultivars, mature plants at the same reproductive stage (full blooming) grown at the same conditions were selected for experimental work in middle to late October 2017.

### 3.2. Root Sample Preparation and Organic Extraction

For each plant, the entire tap root system was collected in the field and transported to the lab. Fresh root tissue was ground into powder in liquid nitrogen. Ethyl acetate (Macklin Technology, Shanghai, China) was added to the powder in a 5:1 (volume to weight) ratio, with nonyl acetate (CAS:143-13-5, ≥98%, Sigma Aldrich, St Louis, MO, USA) included (0.002%) as an internal standard. After shaking at 200 rpm at room temperature for two hours and centrifugation at 5000 rpm for five minutes, the organic phase was collected for subsequent chemical analysis and bioassays.

### 3.3. GC-MS Analysis

A GC-MS system (Agilent Intuvo 9000 GC system coupled with an Agilent 7000D Triple Quadrupole mass detector) was employed for chemical analysis of root extracts as previously reported [10]. Separation was performed on an Agilent HP 5 MS capillary column (30 m × 0.25 mm) with helium as carrier gas (1 mL·min^−1^ of flow rate). The injection volume of each sample was 1 µL. The temperature of the injection port was 260 °C, with a split mode (split ratio = 5:1). The column temperature program of gradient heating was adopted as follows: the temperature was initiated at 40 °C, followed by an increase to 250 °C at a rate of 5 °C/min. The MS conditions included an EI ion source temperature of 230 °C, an ionization energy of 70 eV, and a mass scan range of 40–500 amu. The separated constituents were identified by either using the NIST17 MS library (National Institute of Standards and Technology) or by comparing their mass spectra and retention time with those of the available authentic standards purchased from Sigma Aldrich (St Louis, MO, USA) or obtained from Wilfried A. König. Retention indices were calculated using a series of C7 to C40 hydrocarbon standards (Sigma-Aldrich, St Louis, MO, USA). Each constituent was quantified based on the comparison of its peak area with that of the internal standard, and the contents were expressed as μg g^−1^ fresh weight.

### 3.4. Assessment of Antifungal Activity

Three species of pathogenic fungi, *Magnaporthe oryzae*, *Verticillium dahliae*, and *Fusarium oxysporum*, were tested with the root extracts of 12 cultivars of *C. morifolium* using a mycelial growth assay as previously described [10,23,24]. Fungi cultures were maintained on potato dextrose agar (PDA) medium. To start the mycelial growth assays, a freshly prepared PDA plate (petri dishes of 90 mm in diameter) evenly covered with 200 μL of ethyl acetate root extract of individual cultivars was introduced with a 6 mm plug of mycelial agar of each fungus. After five days of incubation at 28 °C, the diameters of colony zones were measured. The organic solvent ethyl acetate was tested as a negative control. All bioassays were performed in three biological replicates with each replicate containing 10 PDA plates.

### 3.5. Statistical Analysis

Differences in terpenoid concentrations and antifungal activities between root extracts from the 12 cultivars were analyzed by one-way analysis of variance (ANOVA). The analyses were conducted with SAS V8 software (Version 8.02. SAS Institute, Cary, NC, USA) and all statistical effects were considered significant at *p* < 0.05. Principal component analysis (PCA) based on the contents of the terpenoid constituents was performed to classify the tested cultivars. Partial least squares regression analysis (PLS) was performed to assess the relationship between individual terpenoid compounds and antifungal effect. Prior to PLS model processing, data transformations were performed. Colony diameters of the three fungi treated with the extracts of the 12 cultivars of chrysanthemum were divided by the colony diameter of the same fungal species treated with negative control. Then Lg was taken for this value to obtain the final processed data. Next, the loading plot of PLS was conducted using SIMCA-14.1 (Umetrics, Sweden). The contents of all 20 terpenoid compounds were selected for the resulting model of plot loadings, which was treated as the independent variables (X matrix). The transformed values of colony diameters of the three fungi were used as the dependent variables (Y matrix). Then the missing value tolerance (variables set 50% and observations set 50%) and type of scale (Ctr) were set. The root mean square error of estimation (RMSEE) indicated the fit of the observations to the model. The root mean square error of cross validation (RMSECV) was estimated using the cross-validation procedure. The values of R^2^ and Q^2^, which indicate the goodness of fit and the internal predictivity of the PLS models, respectively, were extracted and reported. Permutations analyses were also performed to help assess the risk that the current PLS model is spurious. We selected terpenoid compounds whose variable importance for the projection (VIP)-values larger than 1 (VIP-values larger than 1 indicate “important” X-variables) to indicate the significant correlation with Y-variables.

## 4. Conclusions

Here, we have demonstrated that the roots of all 12 cultivars of *C. morifolium* analyzed contain terpenoids (Figure 1). Sesquiterpenes are the most diverse class, ranging from six to 14 compounds in individual cultivars. In most cultivars, they were also the class of highest concentrations. There was only one diterpene identified, which occurred ubiquitously. In CmR10, the concentration of the diterpene was higher than that of the sesquiterpenes (Figure 2C). No monoterpenes were detected from five cultivars (Figure 1). Considering the long, complex breeding history of Chrysanthemum with thousands of cultivars, the genetic background and the relationships of the cultivars studied in this work are not clear. However, it would be interesting to explore the possible genetic basis underlying the terpenoid composition variability detected in the roots in the future. Terpene synthases are key enzymes for terpene biosynthesis. It is well known that a few amino acid changes in terpene synthases may lead to distinct product profiles [25,26,27]. It is tempting to speculate that sequence variations among alleles of terpene synthase genes are responsible for the observed terpene profiles among roots of the 12 cultivars. It is equally interesting to ask what the biological/ecological significance of such cultivar-specificity of terpenes is. As introduced previously, root secondary metabolites, including terpenes, are implicated in diverse interactions between roots and other organisms in the rhizosphere [28,29,30]. It is certainly sensible to hypothesize that root terpenes of *C. morifolium* act as a chemical defense against various pathogenic microbes and insects. Our evaluation of their effects on fungal growth supports such a hypothesis (Figure 4). There are two observations. One is that root extracts of *C. morifolium* are generally less effective in inhibiting the growth of *F. oxysporum* than *M. oryzae* and *V. dahliae* (Figure 4). Second is that there are variations among the 12 cultivars in inhibiting the growth of a specific fungus. Such variations may suggest the adaptive nature of such terpenoids, as different cultivars may be adapted to different environments with different pest pressure [31]. This work may inspire a number of interesting future studies. For example, root extracts of certain cultivars of *C. morifolium* could be further explored as an antifungal agent with thorough comparison to commercial products. The effects of individual terpenoids on inhibiting fungal growth are generally inconclusive at this time due to the statistically poor models generated from PLS analysis (Figure 5). Nonetheless, it will be interesting to further evaluate the antifungal effects of terpenoid compounds individually and in combinations. While this study focuses on terpenoids, it will also be interesting to determine whether non-terpenoid compounds (Appendix A) and polar components from roots of *C. morifolium* have antifungal properties. In brief, our study provides novel information about the chemical composition of secondary metabolites and their function from the roots of *C. morifolium*, which may open up new exciting questions about the biosynthesis, regulation, and biological functions of these secondary metabolites.

## Figures and Tables

**Figure 1 molecules-25-02083-f001:**
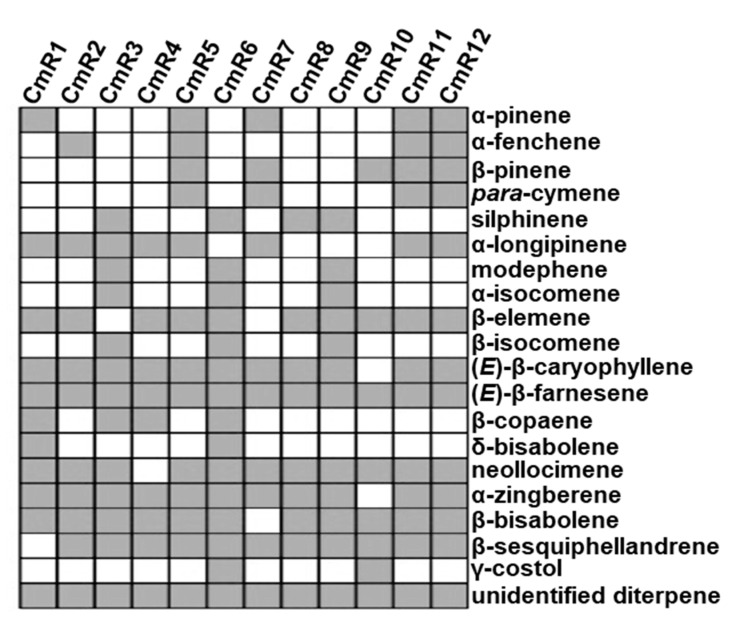
Presence/absence of individual terpenoids in roots of 12 cultivars of *C. morifolium* (CmR1 to CmR12). Gray and white rectangles denote the presence and absence of a terpenoid, respectively. CmR1–12 refer to the cultivar codes in Table 1.

**Figure 2 molecules-25-02083-f002:**
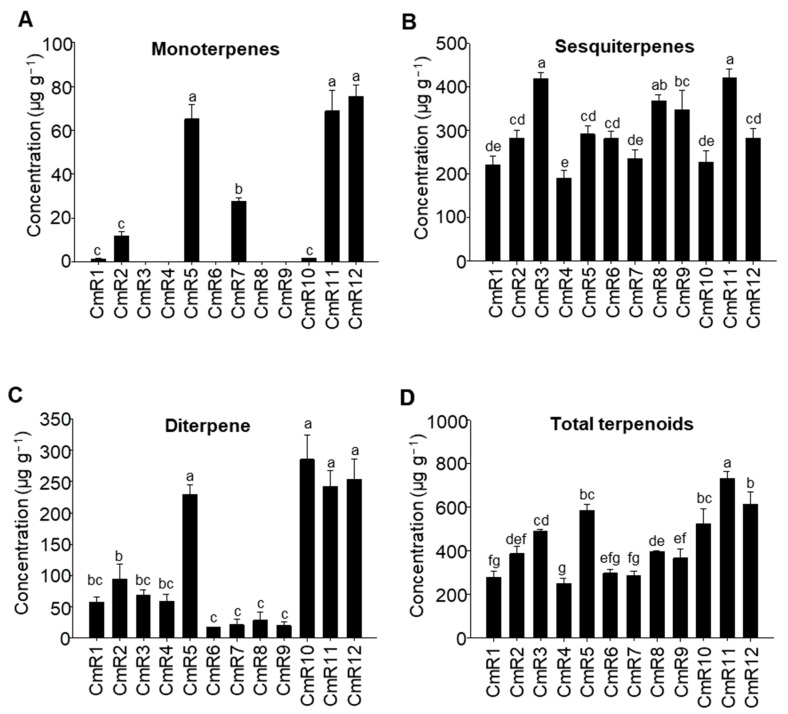
Concentrations of the monoterpenes (**A**), sesquiterpenes (**B**), diterpene (**C**), and total terpenoids (**D**) in root extracts of 12 cultivars of *C. morifolium.* CmR1–12 refer to the cultivar codes in Table 1. Different letters in (**A**–**D**) denote statistically significant differences among the means according to ANOVA analysis (*p* < 0.05).

**Figure 3 molecules-25-02083-f003:**
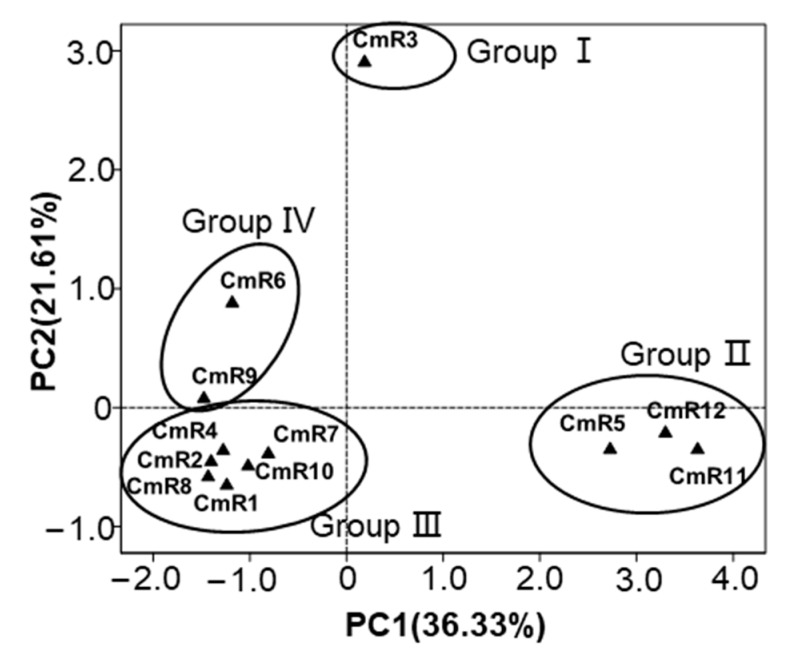
Principal component analysis (PCA) of the detected terpenoids of roots from 12 cultivars of *C. morifolium.* CmR1–12 refer to the cultivar codes in Table 1.

**Figure 4 molecules-25-02083-f004:**
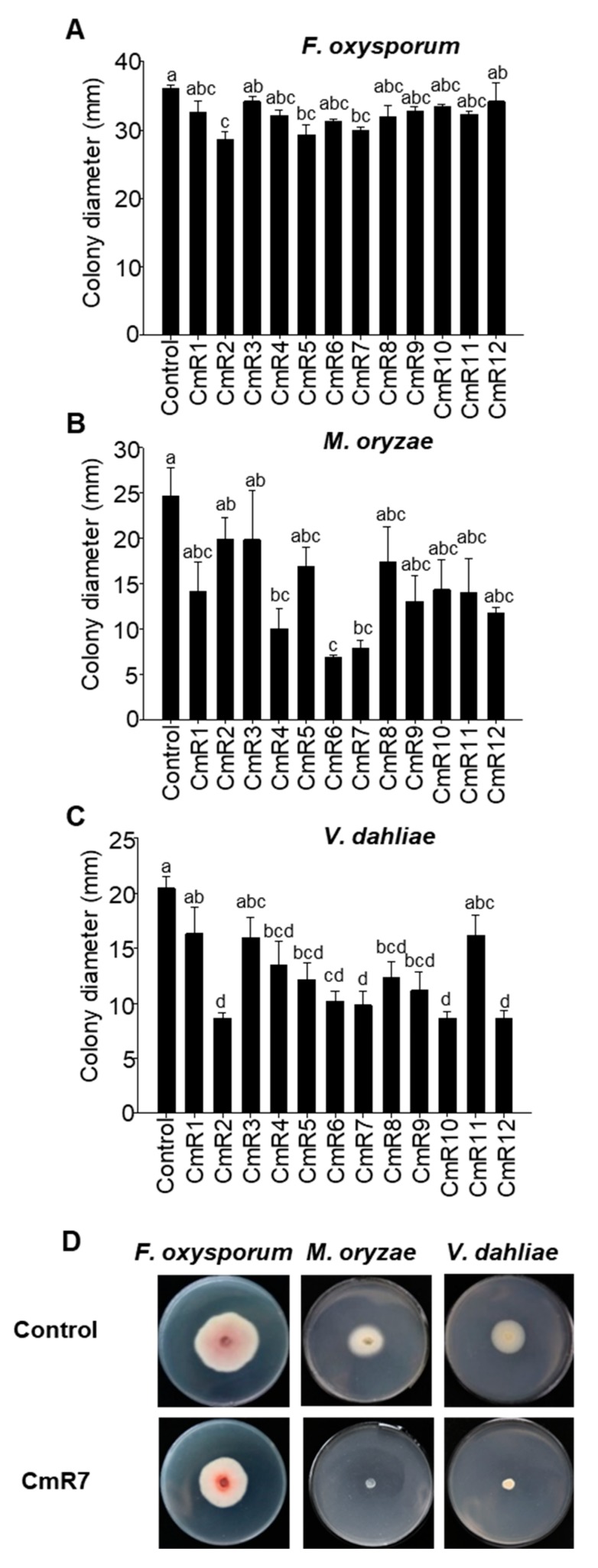
Effect of terpenoids from 12 cultivars of *C. morifolium* on the growth of three pathogenic fungi species: *Fusarium oxysporum* (**A**), *Magnaporthe oryzae* (**B**), and *Verticillium dahliae* (**C**). “Control” depicts a negative control with organic solvent ethyl acetate. (**D**) Representative growth of *F. oxysporum*, *M. oryzae*, and *V. dahliae* treated with a negative control (ethyl acetate) or with an extract made from CmR7 roots. CmR1–12 refer to the cultivar codes in Table 1. Different letters in (**A**–**C**) denote statistically significant differences among the means according to ANOVA analysis (*p* < 0.05).

**Figure 5 molecules-25-02083-f005:**
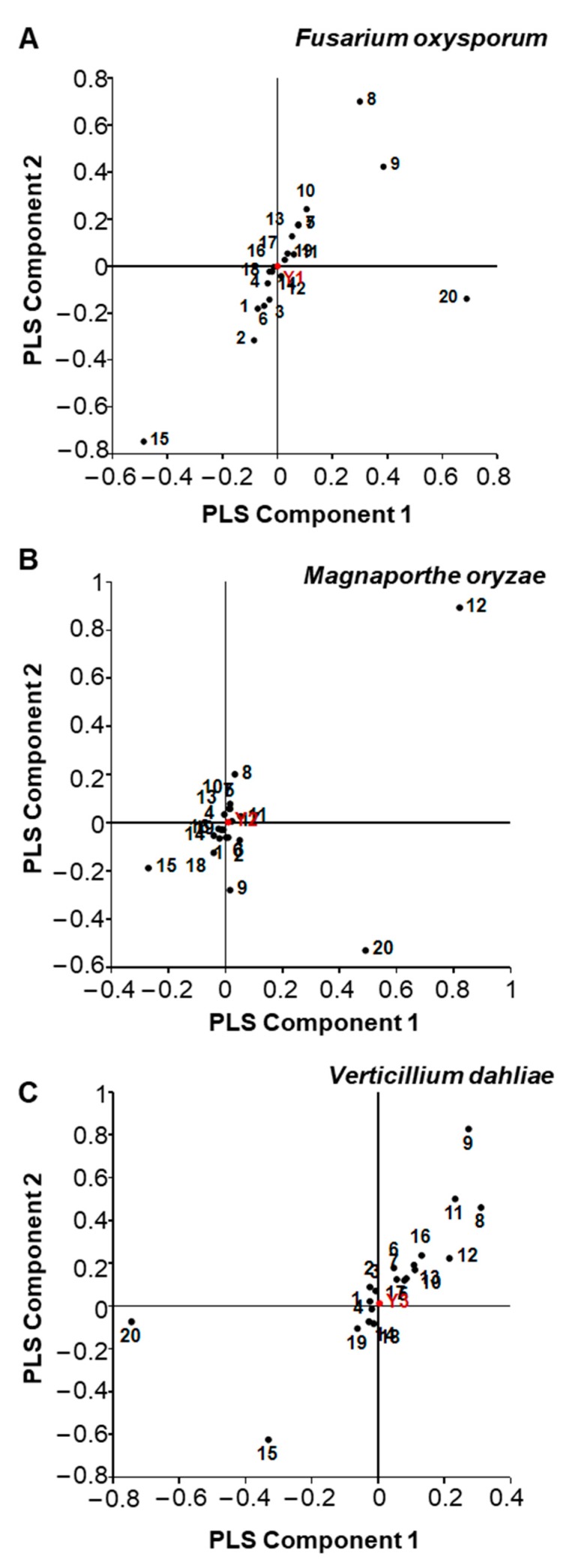
Partial least squares (PLS) loadings for colony diameter (Y-variables) of *Fusarium oxysporum* (**A**), *Magnaporthe oryzae* (**B**), *Verticillium dahliae* (**C**) and the content of terpenoid compounds (X-variables) in 12 cultivars of *C. morifolium*. Number 1–20 refer to the codes of terpenoid compounds in Table 2.

**Table 1 molecules-25-02083-t001:** Cultivars of *C. morifolium* used in this study.

Code	Cultivar	Collection Locality
CmR1	‘Hang Bai Ju’	Nanjing, Jiangsu province, China
CmR2	‘Xiao Xiang Ju’	Nanjing, Jiangsu province, China
CmR3	‘Shi Huang Ju’	Nanjing, Jiangsu province, China
CmR4	‘E Pang Gong’	Nanjing, Jiangsu province, China
CmR5	‘Da Yang Ju’	Nanjing, Jiangsu province, China
CmR6	‘Huang Ju’	Nanjing, Jiangsu province, China
CmR7	‘Xiao Huang Ju’	Nanjing, Jiangsu province, China
CmR8	‘Wan Gong Ju’	Nanjing, Jiangsu province, China
CmR9	‘Xiu Ning Yao Ju’	Nanjing, Jiangsu province, China
CmR10	‘Bai Xiang Li’	Nanjing, Jiangsu province, China
CmR11	‘She Yang Hong Xin Ju’	Nanjing, Jiangsu province, China
CmR12	‘Zao Hua’	Nanjing, Jiangsu province, China

**Table 2 molecules-25-02083-t002:** Terpenoid constituents and concentrations (mean ± SD) from roots of 12 cultivars of *C. morifolium*.

NO.	Compounds	Retention Index	Concentration (μg g−1 Fresh Weight)											
			CmR1	CmR2	CmR3	CmR4	CmR5	CmR6	CmR7	CmR8	CmR9	CmR10	CmR11	CmR12
	**Monoterpenes**													
**1**	α-pinene *	939	1.32 ± 0.15 ^a^	ND	ND	ND	15.34 ± 1.77	ND	15.11 ± 1.73	ND	ND	ND	14.70 ± 2.23	16.90 ± 2.28
**2**	α-fenchene	950	ND ^b^	11.76 ± 1.87	ND	ND	27.64 ± 2.29	ND	ND	ND	ND	ND	32.04 ± 4.12	33.95 ± 2.68
**3**	β-pinene *	981	ND	ND	ND	ND	17.84 ± 2.26	ND	4.03 ± 1.05	ND	ND	1.52 ± 0.30	17.14 ± 2.19	19.45 ± 2.34
**4**	*para*-cymene	1035	ND	ND	ND	ND	4.40 ± 0.32	ND	8.46 ± 1.91	ND	ND	ND	4.95 ± 0.76	5.15 ± 0.61
	**Sesquiterpenes**													
**5**	Silphinene	1353	ND	ND	19.55 ± 0.62	ND	ND	8.52 ± 0.31	ND	1.85 ± 0.15	5.24 ± 0.71	ND	ND	ND
**6**	α-longipinene	1361	2.37 ± 0.15	1.75 ± 0.35	1.71 ± 0.13	2.28 ± 0.36	17.13 ± 1.27	ND	8.14 ± 0.86	ND	ND	ND	23.01 ± 2.76	18.38 ± 1.91
**7**	modephene	1391	ND	ND	20.73 ± 0.51	ND	ND	9.41 ± 0.30	ND	ND	4.99 ± 0.52	ND	ND	ND
**8**	α-isocomene	1396	ND	ND	80.87 ± 2.33	ND	ND	41.03 ± 1.26	ND	ND	22.98 ± 2.99	ND	ND	ND
**9**	β-elemene	1399	53.40 ± 7.66	2.01 ± 0.47	ND	47.12 ± 2.24	55.20 ± 3.01	12.00 ± 1.48	ND	2.38 ± 0.65	45.63 ± 5.22	81.06 ± 14.22	84.53 ± 6.83	42.40 ± 0.83
**10**	β-isocomene	1418	ND	ND	27.87 ± 0.76	ND	ND	12.50 ± 0.44	ND	ND	7.54 ± 0.92	ND	ND	ND
**11**	(*E*)-β-caryophyllene *	1431	10.93 ± 1.64	10.00 ± 0.80	35.58 ± 0.56	7.86 ± 1.13	27.12 ± 0.39	18.05 ± 0.90	6.93 ± 0.40	11.50 ± 0.56	11.15 ± 0.67	ND	37.53 ± 2.77	32.12 ± 4.45
**12**	(*E*)-β-farnesene	1464	112.23 ± 8.31	246.48 ± 17.39	175.81 ± 12.22	108.22 ± 11.61	131.77 ± 12.88	68.36 ± 6.21	66.56 ± 8.69	258.40 ± 8.14	207.81 ± 30.94	127.94 ± 11.08	194.25 ± 9.79	133.47 ± 9.01
**13**	β-copaene	1474	5.80 ± 1.64	ND	12.74 ± 0.37	7.56 ± 0.85	ND	6.85 ± 0.93	ND	ND	ND	ND	ND	ND
**14**	δ-bisabolene	1489	2.54 ± 0.77	ND	ND	ND	ND	14.85 ± 1.17	ND	ND	ND	ND	ND	ND
**15**	Neollocimene	1495	5.11 ± 0.55	6.81 ± 1.21	11.67 ± 0.28	ND	15.97 ± 0.61	9.99 ± 0.54	133.61 ± 14.34	11.46 ± 1.98	4.26 ± 1.20	4.31 ± 0.38	18.18 ± 1.04	16.40 ± 2.53
**16**	α-zingberene	1504	22.95 ± 2.12	3.56 ± 0.80	5.49 ± 0.80	6.27 ± 1.32	13.19 ± 1.02	20.38 ± 0.97	7.77 ± 1.16	18.88 ± 0.70	10.62 ± 0.47	ND	17.13 ± 1.30	10.44 ± 1.45
**17**	β-bisabolene *	1518	5.43 ± 0.78	4.84 ± 0.58	8.25 ± 1.63	6.45 ± 0.89	9.50 ± 0.35	5.76 ± 0.73	ND	9.81 ± 0.22	3.77 ± 0.74	4.52 ± 0.76	11.49 ± 0.88	10.53 ± 0.82
**18**	β-sesquiphellandrene	1533	ND	6.08 ± 0.87	19.05 ± 2.05	4.95 ± 0.73	21.17 ± 2.15	49.39 ± 2.85	12.45 ± 2.40	53.40 ± 5.34	23.34 ± 2.27	9.11 ± 0.57	34.19 ± 4.01	18.65 ± 2.96
**19**	γ-costol	1760	ND	ND	ND	ND	ND	3.91 ± 1.34	ND	ND	ND	10.84 ± 0.67	ND	ND
	**Diterpene**													
**20**	unidentified diterpene	2298	57.39 ± 7.58	93.93 ± 23.83	69.13 ± 8.21	58.66 ± 11.25	229.37 ± 16.19	16.99 ± 0.64	20.94 ± 8.79	28.38 ± 12.90	19.58 ± 5.54	285.52 ± 39.55	242.43 ± 24.56	253.68 ± 32.81

^a^ Mean values were calculated from three biological replicates. ^b^ Not detected. * identified by authentic standard.

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
