# Peer review of "Diverse Terpenoids and Their Associated Antifungal Properties from Roots of Different Cultivars of Chrysanthemum Morifolium Ramat"

_molecules, 2020, doi:10.3390/molecules25092083_

Round 1
Reviewer 1 Report
The manuscript presented reports the findings of a study aimed to determine the relationship between the chemical composition of root extracts of 12 cultivars of Chrysanthemum morifolium and their antifungal activity against Fusarium oxysporum, Magnaporthe oryzae, and Verticillium dahlia. The relationship was determined by using Principal Component Analysis and Hierarchical Cluster Analysis.
In general terms, the manuscript is well-written. The experimental results are well-presented and properly discussed. For these reasons, I am principally positive towards the publication of this work in Molecules.
However, minor revision and some corrections should be done first:
1.- During the discussion of results the authors mentioned the cultivars by their proper name and in the parentheses provide their corresponding code. It was quite confusing and made difficult the reading, thus I suggest keep the codes and eliminated the proper names.
2.- It is neccessary that the authors menttined which was the positive control used.
3.- The authors did not provide information about the genetic profile of the investigated cultivars. I think it is necessary to include this information in the Introduction section as well as in the discussion of the results. Moreover, take into account that the cultivars were collected in the same locality, I consider it would be beneficial to present a discussion about how the terpenoid composition variability detected in the roots of the investigated cultivars could be linked to their genetic profile.
Author Response
Review 1:
The manuscript presented reports the findings of a study aimed to determine the relationship between the chemical composition of root extracts of 12 cultivars of Chrysanthemum morifolium and their antifungal activity against Fusarium oxysporum, Magnaporthe oryzae, and Verticillium dahlia. The relationship was determined by using Principal Component Analysis and Hierarchical Cluster Analysis.
In general terms, the manuscript is well-written. The experimental results are well-presented and properly discussed. For these reasons, I am principally positive towards the publication of this work in Molecules.
We thank the reviewer for this positive evaluation!
However, minor revision and some corrections should be done first:
1.- During the discussion of results the authors mentioned the cultivars by their proper name and in the parentheses provide their corresponding code. It was quite confusing and made difficult the reading, thus I suggest keep the codes and eliminated the proper names.
Response: The codes were kept and the real names were eliminated as suggested by the reviewer.
2.- It is necessary that the authors mentioned which was the positive control used.
Response: The objective of this study is to compare the differences in chemical composition of root secondary metabolites and their antifungal properties. It was not our aim to compare Chrysanthemum root extracts to commercial fungicides, which could be an entirely new, stand-alone study. Therefore, we do not have a selected positive control for our experiments, but have used a proper negative control.
The authors did not provide information about the genetic profile of the investigated cultivars. I think it is necessary to include this information in the Introduction section as well as in the discussion of the results. Moreover, take into account that the cultivars were collected in the same locality, I consider it would be beneficial to present a discussion about how the terpenoid composition variability detected in the roots of the investigated cultivars could be linked to their genetic profile.
Response: Due to the long, complex breeding history of Chrysanthemum with thousands of cultivars, the genetic background of the cultivars studied in this work and their relationships are unknown. But as suggested by the reviewer, we have now added additional discussion about the possible genetic basis underlying the terpenoid composition variability detected in the roots.
Reviewer 2 Report
The authors describe GC-MS analyses of 12 cultivars of Chrysanthemum morifolium Ramat roots. The antifungal properties of extracts were also evaluated.
Major comment
・There has been no report on the difference in the composition of this plant roots cultivars for GC-MS analysis. However, it seems insufficiency since analyses in this manuscript are limited to low-polarity and low-molecular compounds. The results of analyses in this paper are related to the antifungal properties of extracts. However, medium polar and highly polar components should also be analyzed, and their results should be considered together for antifungal properties.
Minor comment
・Authors should identify an unidentified diterpene.
・Authors should show GC-MS chromatograms.
Author Response
Review 2:
The authors describe GC-MS analyses of 12 cultivars of Chrysanthemum morifolium Ramat roots. The antifungal properties of extracts were also evaluated.
Major comment:
There has been no report on the difference in the composition of this plant roots cultivars for GC-MS analysis. However, it seems insufficiency since analyses in this manuscript are limited to low-polarity and low-molecular compounds. The results of analyses in this paper are related to the antifungal properties of extracts. However, medium polar and highly polar components should also be analyzed, and their results should be considered together for antifungal properties.
Response: We agree that metabolites of other chemical classes from Chrysanthemum may also have antifungal effects. However, measuring and analyzing more polar compounds by e.g. LC-MS approaches is beyond the scope of this work, but can be an interesting goal for future studies.
Minor comment:
・Authors should identify an unidentified diterpene.
Response: Structural elucidation of the diterpene as requested by the reviewer is a very hard and long-term task that needs preparative GC and subsequent NMR analysis. Although the reviewer is right that such elucidation is worthwhile, we feel that it is beyond the scope of our study. Nonetheless, to make the spectral data accessible for interested readers, we now present the mass spectrum of the unknown diterpene in supplemental figure S1.
・Authors should show GC-MS chromatograms.
Response: Because we analyzed a larger number of cultivars with biological replicates, the total number of obtained GC chromatograms is very high. Thus we originally decided to organize and show the GC results in a table in the main manuscript (Table 2). However, to give the reader an impression about the chromatograms, we now present one representative GC chromatogram for CmR6, which have most diverse profile of terpenoids, in the new supplemental figure S2.
Round 2
Reviewer 2 Report
I think there is little improvement in the manuscript.
・At least the relatively large peaks appearing after 30 min in the GC-MS chromatogram in SI should be identified.
・SI should include all GC-MS chromatograms.
I cannot recommend accepting this manuscript for publication.
Author Response
Reviewer 2
・At least the relatively large peaks appearing after 30 min in the GC-MS chromatogram in SI should be identified.
The large peak in the GC chromatogram as supplementary figure 2 is tentatively identified. This information has been added to the revised figure S2 and properly described in the main text. In addition, some additional discussions were added to the main text in regards to the non-terpenoid metabolites.
・SI should include all GC-MS chromatograms.
We respectfully disagree with this request. The data based on GC-MS analysis have been extracted and organized in Table 2. It would be redundant to present both GC chromatograms and the table. We elected to present the table, which give more clear information for reader to follow.